# On-chip phonon-magnon reservoir for neuromorphic computing

Dmytro D. Yaremkevich[1], Alexey V. Scherbakov [1] ✉, Luke De Clerk [2,3], Serhii M. Kukhtaruk [4], Achim Nadzeyka [5], Richard Campion[6], Andrew W. Rushforth [6], Sergey Savel'ev [2], Alexander G. Balanov [2] & Manfred Bayer [1]

Reservoir computing is a concept involving mapping signals onto a high-dimensional phase space of a dynamical system called "reservoir" for subsequent recognition by an artificial neural network. We implement this concept in a nanodevice consisting of a sandwich of a semiconductor phonon waveguide and a patterned ferromagnetic layer. A pulsed write-laser encodes input signals into propagating phonon wavepackets, interacting with ferromagnetic magnons. The second laser reads the output signal reflecting a phase-sensitive mix of phonon and magnon modes, whose content is highly sensitive to the write- and read-laser positions. The reservoir efficiently separates the visual shapes drawn by the write-laser beam on the nanodevice surface in an area with a size comparable to a single pixel of a modern digital camera. Our finding suggests the phonon-magnon interaction as a promising hardware basis for realizing on-chip reservoir computing in future neuromorphic architectures.

The flourishing idea of using artificial neural networks (ANNs) for the solution of practical tasks shifts the computing paradigm towards heavy usage of data instead of creating lengthy instructions and logical chains forming algorithms. Being inspired by biological neural systems, an ANN constitutes a set of many interconnected non-linear functional units called 'neurons'. To solve a task with the desired accuracy, an ANN requires training by data processing to fit the weights of the connections between neurons. Over the decades, ANNs have demonstrated their high efficiency in a wide range of important practical applications[1] including voice, image, and pattern recognition, data mining, prediction of complex dynamics, 3D reconstruction, and medical diagnostics.

The more sophisticated the problem an ANN has to solve, the larger is the network required to achieve the desirable accuracy of the solution. Simulation of large networks using conventional digital computers demands enormous CPU/GPU power, memory, and energy supply[2]. A promising approach to resolve the hardware issues is neuromorphic computing, which implements ANNs in circuits[3]. Such hardware architecture should dramatically improve the transfer of data across the circuit during data processing, thus enhancing both the speed and energy efficiency of calculations.

Another common drawback of ANNs is the significant time required for their training to optimize performance[4]. ANNs are most efficient if the type of task ('the goal') is fixed, and there is enough data for training. However, they experience significant difficulties if the amount of data is limited, or the goal varies in time[5]. Smaller networks would partly resolve this problem since fewer neurons and connections need less data and time for training, but this approach suffers from poor accuracy. A promising way to resolve this problem is the concept of 'reservoir computing' (RC). The 'reservoir' is a recurrent ANN with non-linear dynamics and fixed connections, which typically does not require any training, attached to a small readout ANN, which

[1]Experimentelle Physik 2, Technische Universität Dortmund, D-44227 Dortmund, Germany. [2]Department of Physics, Loughborough University, Loughborough LE11 3TU, UK. [3]Machine Learning Development, SS&C Technologies, 128 Queen Victoria Street, London EC4V 4BJ, UK. [4]Department of Theoretical Physics, V. E. Lashkaryov Institute of Semiconductor Physics, 03028 Kyiv, Ukraine. [5]Raith GmbH, 44263 Dortmund, Germany. [6]School of Physics and Astronomy, University of Nottingham, Nottingham NG7 2RD, UK. ✉e-mail: alexey.shcherbakov@tu-dortmund.de

is trained to recognize predefined characteristics of the reservoir output. In such an architecture, the reservoir serves to 'highlight' important features in data sets and suppress irrelevant ones, thereby enormously improving the performance of computing.

As one of many natural prototypes of RC, one could consider the human vision schematically illustrated in Fig. 1a. The visual information passed through the cornea and focused by the lens is converted by the retina photoreceptors into electrical signals (the neural pulses) that are transmitted by the optic nerve and mixed in a high-dimensional space of the visual cortex for the subsequent recognition. The complex photochemical transformation (phototransduction) by the $10^8$ photoreceptors allows our brain to recognize objects, distinguish tiny differences between them, and detect thereby their motion. Such a resemblance between biological sensory systems and reservoir ANNs has invigorated research on RC-based artificial vision[6], and other RC-systems for detection and recognition[5,7].

From the viewpoint of information theory, the reservoir maps input signals into higher dimensional information spaces and can be realized by continuous in space and/or in time nonlinear dynamical systems with storage ability. Therefore, various physical systems have been suggested for the reservoir[5,8–10]. One of the first realizations was based on waves, namely gravity waves in a vessel with water to recognize spoken numbers[11]. Nowadays, wave 'reservoirs' have been implemented using different physical waves such as electromagnetic waves (photons)[12–18], elastic waves (phonons)[19,20], and spin waves (magnons)[21–24]. The key factors that make waves reliable for reservoir architectures are the possibility to achieve wave packets with a vast information density; weighted summation by interference;

nonlinearity resulting in wave mixing and generation of higher harmonics; hybridization of waves of different types, e.g., the formation of polarons. Recently, wave-based 'reservoirs' have demonstrated the ability for deep learning[25–27].

Despite rapid progress in the development of RC-systems, their implementation in chips suitable for practical technological applications has remained an unsolved problem. Among the major challenges with existing neuromorphic platforms[3] are power efficiency, miniaturization, separability, and robustness[5,8,10]. On-chip wave-based reservoirs have potential to address these challenges, since they allow smaller devices, good scalability, great state numbers, and fine control of interacting wave modes. However, their development calls for novel materials and designs allowing the generation of waves for the reservoir function.

## Results

### Hybrid nanostructure for phonon–magnon reservoir

Here, we propose and demonstrate experimentally a new type of wave reservoir based on high-frequency acoustic and spin waves in a physical system that combines ultimate compactness, robustness, and separability. For this aim, we fabricate a nanodevice with an architecture implemented with available nanotechnology. The device, shown schematically in Fig. 1b, consists of a sandwich of GaAs/AlAs semiconductor layers, which form a waveguide for high-frequency acoustic waves, i.e., phonons[28]. The waveguide is capped by a nano-grating patterned in a metal ferromagnetic $Fe_{0.81}Ga_{0.19}$ layer hosting spin waves, i.e. magnons[29]. Figure 1c shows the scanning electron microscopy image of the patterned surface. We examine the

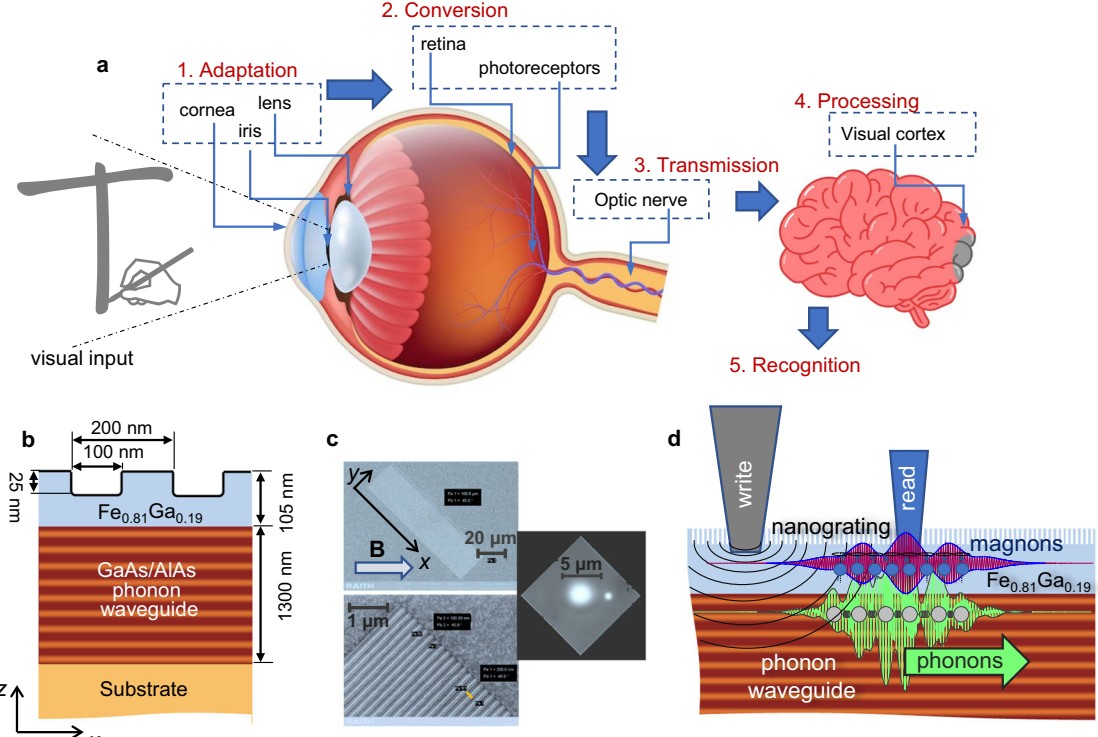

**Fig. 1 | Nanodevice architecture and working principle. a** Schematic of human vision: an image focused by the lens onto the retina is transformed into electrical pulses that are transferred by the optic nerve to the brain for the subsequent recognition. **b** Design of the examined nanostructure. **c** Scanning electron microscopy (SEM) images (top and bottom panels) of the nanodevice surface with the nanograting (NG): complete NG with a size of $25 \times 100\ \mu m^2$ (top panel), and its close-up with the visible pattern (bottom panel). The inset in the right-top corner is a microscopy image of the write (large spot) and read (small spot) laser spots on the

NG surface. The size of the framed area is $10 \times 10\ \mu m^2$. **d** Working principle of the phonon reservoir with magnon readout. A multimode phonon wavepacket is generated by ultrafast laser excitation of the patterned ferromagnetic nanolayer. It propagates well protected from external perturbations beneath the surface in the GaAs/AlAs phonon waveguide and its continuously transformed waveform is imprinted in the magnon modes of the ferromagnetic layer. The superposition of the magnon modes driven by the phonon wavepacket at a specific distance from the write spot is detected using the read laser pulse.

performance of this device by recognition of the "visual shapes" drawn on the surface by a pulsed 'write' laser. The laser pulses are converted by the nanograting to a coherent multimode phonon wavepacket, which is guided along the surface and experiences continuous transformation due to the interference of the phonon modes. Phonons interact with magnons and this multimode magnon-phonon mixture forms the reservoir multidimensional space. The reservoir output is read through magnons by measuring the reflectivity of another 'read' laser beam. The output waveform is integrated over the drawn trajectory corresponding to a specific "shape" for the following recognition by an ANN programmed in a personal computer. We may compare this procedure with handwriting and conceptualize the reservoir operations as recognition of a handwritten character. Thus, in analogy with the human vision (Fig. 1d), we may consider the patterned surface as the retina, which converts optical input to neural pulses (coherent phonons), processed by a multidimensional reservoir space (phonon-magnon mixture) to generate a readout recognized by the brain, namely the ANN. The sub-μm wavelength of phonons and magnons allows achieving a large number of "receptors" per area ("receptor"

density) and make the readout sensitive to a tiny change of the optical input. It enables recognition of the symbols drawn on the scale of just several wavelengths of the 'write' laser.

The physical processes involved in the operation of the fabricated nanodevice are demonstrated in Fig. 2. The 100-fs pulse of the write laser, focused on the metallic ferromagnetic layer to a spot of 2-μm diameter instantaneously induces thermal stress leading to the generation of the coherent acoustic wavepacket of hypersound (up to ~100 GHz) frequencies[30,31]. Due to the periodic surface patterning and the corresponding spatial modulation of the optically induced stress along the $x$-axis, coherent phonons acquire the wavevectors: $k_x = n\,2\pi/d$, where $d = 200$ nm is the nanograting period and $n = 1, 2, 3...$ is the harmonic order[32]. The phonon wavepacket propagates along the $x$-axis away from the optically excited area. There are two groups of propagating phonon modes with alternative localizations. The surface modes of two polarizations, so-called Rayleigh and Sezawa waves, with the frequencies of 12.2 and 13.8 GHz ($n = 1$), respectively, are localized in the near-surface region. Their propagation is suppressed due to the scattering at the corrugated surface, and their amplitudes drastically

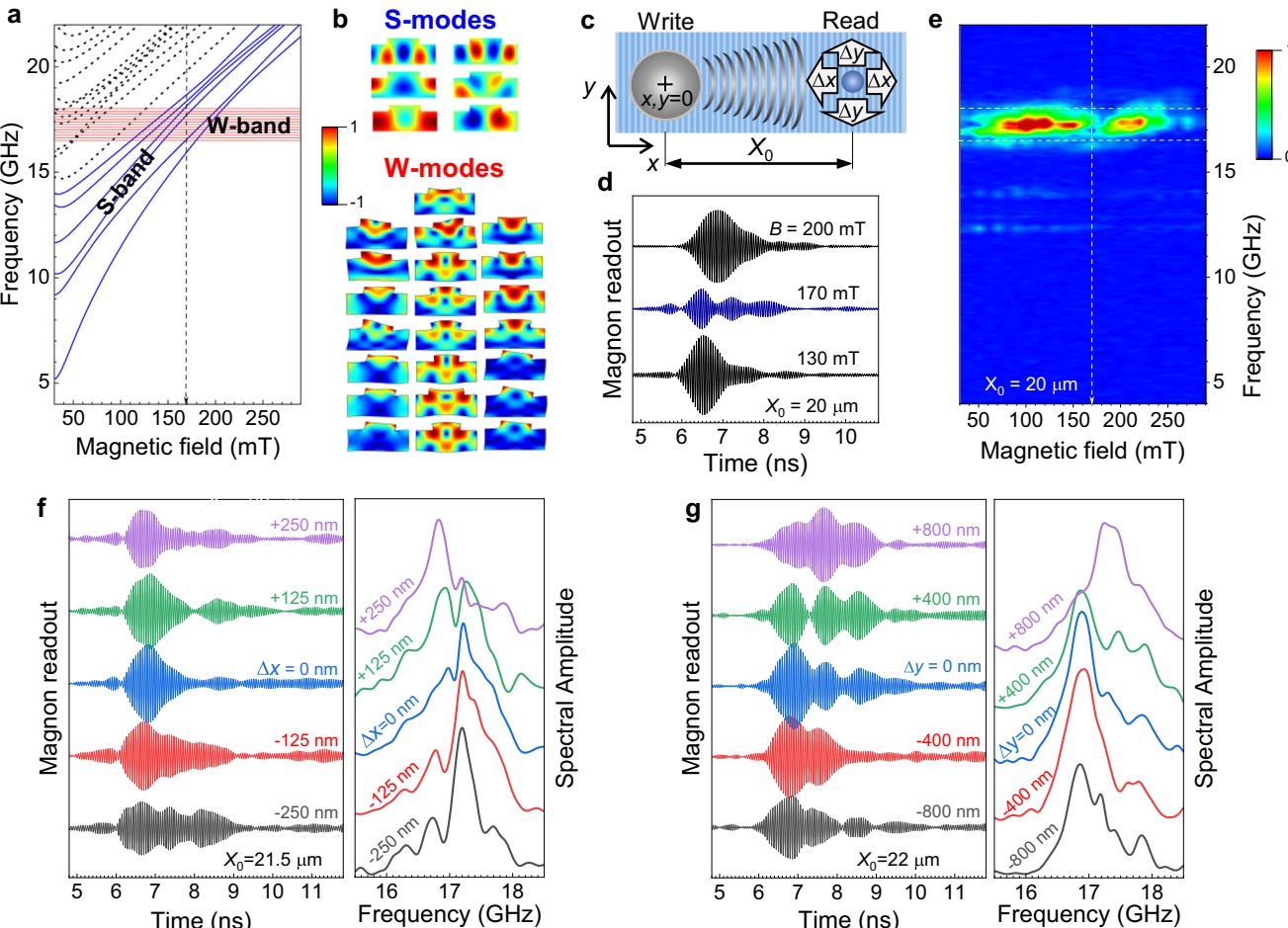

**Fig. 2 | Physical principle of the phonon-magnon reservoir. a** Calculated dependences of the frequencies of the nanograting magnon modes on magnetic field. Blue solid lines show the 6 lowest modes (forming the S-band). The external magnetic field, **B**, is applied in the layer plane at 45° angle relative to the NG edge. The spectral range of the 1st order W-band is shown by the pink dashed rectangle. The vertical dashed line shows the magnetic field strength at which the spectral centres of the S- and W-bands coincide. **b** Calculated spatial profiles of the 22 W-modes (normalized absolute value of displacement vector, exaggerated for clarity) and 6 lowest S-modes in the ferromagnetic layer ($z$-projection of the normalized magnetization). **c** Coordinate system and relative positions of the write and read spots in the experiments. **d** Noise-free readout signals measured for three

values of $B$ at 20-μm distance between the write and read spots. **e** Colour map showing the field dependence of the spectral amplitude obtained by a fast Fourier transform of the readout signals measured at $X_0 = 20$ μm. The vertical dashed line shows the magnetic field strength, at which the magnon readout demonstrates the most complex waveform. The horizontal dashed lines indicate the calculated spectral position of the guided phonon W-band. **f, g** Typical magnon noise-free readouts measured at various horizontal (**f**) and vertical (**g**) relative shifts of the write and read spots. Left panels show the transient signals, right panels the respective fast Fourier transforms. Time $t = 0$ corresponds to the moment when the input laser pulse hits the ferromagnetic layer.

decrease within the propagation distances of several microns. The second group of modes, so-called W-modes, are localized in the GaAs/AlAs sandwich, which has a higher acoustic impedance than the impedances of the substrate and the ferromagnetic cap. The W-modes are analogue to the Lamb modes in plates and behave as waveguide modes, which can propagate on large distances protected from scattering[28,33]. Detailed information about the spectrum of the optically generated coherent phonons is presented in Supplementary Fig. 1. Here we focus on the W-modes, which are the main information carriers in the proposed device. The number of W-modes for a specific wavevector is determined by its ratio to the total waveguide thickness. Numerical modelling (Methods) shows that the first-order W-band ($n = 1$) consists of 22 W-modes with frequencies between 16.5 and 18 GHz as shown by the red dashed rectangle in Fig. 2a. The spectral bands with higher order $n$ (not shown) contain a continuously increasing number of phonon modes.

Due to the magneto-elastic coupling the propagating phonon wavepacket interacts with magnons in the ferromagnetic layer[34,35]. The magnon spectrum includes a number of spectrally close S-modes formed by spin waves spatially modulated along the $x$-axis and quantized along the normal to the layer plane ($z$-axis)[29,36]. The frequencies of the magnon modes depend on the strength of the external magnetic field, $B$, applied in our experiments in the layer plane at an angle of 45 degrees relative to the direction of the nanograting grooves. Figure 2a shows the calculated frequencies of the magnon modes (blue and dashed black lines) as a function of $B$. For any pair of phonon and magnon modes, the efficiency of their interaction depends on their spatial and spectral matching[37,38]. The spatial matching is determined by the mode profiles set by the device structural design. Figure 2b shows the calculated spatial profiles of the 22 first-order W-modes and the six lowest S- modes in the ferromagnetic layer. Their interaction occurs through the uniaxial ($xx$) and shear ($xz$) strain components of the W-modes[37] and for any pair of the illustrated W- and S-modes the overlap integral is nonzero. The spectral matching is set by the external magnetic field. The highest interaction efficiency takes place at the resonance condition when the frequencies of the interacting modes are equal. However, due to the finite spectral widths of the S- and W-modes, their interaction is still quite efficient when the modes are detuned. Moreover, magnon-phonon interaction is intrinsically nonlinear and is accompanied by the parametric effects, which result in frequency mixing of the phonon modes[39]. As a result, the magnon readout signal, $M(t)$, is characterized by a complex waveform, which is the phase-sensitive superposition of all S- and W-modes in the detection area, converted to an electric signal exploiting the polar magneto-optical Kerr effect (Methods).

The noise-free waveforms (see Methods) of the magnon readouts, $M(t)$, measured at three values of $B$ at the distance $X_0 = 20 \, \mu m$ between the write and read laser spots are shown in Fig. 2d. The most complex readout is observed at $B = 170 \, mT$, which corresponds to the intersection of the W-band and the six lowest S-modes, which profiles are shown in Fig. 2b. The readout complexity manifests in the maximal broadening of its fast Fourier transform spectrum, the field dependence of which is shown in Fig. 2e. The experimental results demonstrated further correspond to these experimental conditions.

Figure 2f, g demonstrate how the magnon readout $M(t)$ depends on the position of the read point relative to the write point where the phonon wavepacket is generated (for the scheme, see Fig. 2c). A tiny change in the relative positions of the write and read points, separated by a distance of more than $20 \, \mu m$, results in a noticeable modification of the readout waveform and its spectrum. We have checked that the device is homogeneous, and the readout signal $M(t)$ does not depend on the position at the device surface when the relative position of the input and output spots is preserved. A 125-nm step in the horizontal direction is smaller than the nanograting period (the W- and S-modes' wavelength), and much smaller than the write and read laser

wavelengths of 1050 and 780 nm as well as spots diameters of 2 and 0.6 μm, respectively. The readout waveform is sensitive to both horizontal and vertical shifts, $\Delta x$ and $\Delta y$. The strong sensitivity of $M(t)$ to the relative position is governed by the properties of the propagating phonon wavepacket. Each W-mode is characterized by its individual spatial profile, frequency, velocity, and decay rate and excited by the write laser pulse with individual initial amplitude and phase. As a result of the modes' interference, the phonon wavepacket experiences continuous transformation during its propagation along the $x$-axis as illustrated in the Supplementary Video. Moreover, the superposition of the phonon modes varies also along the wavefront ($y$-axis) due to the finite sizes of the write and read laser spots. This spatial-temporal transformation imprinted into the magnon S-modes results in a unique spatial, temporal, and spectral variation of the readout. Notably, it is reminiscent of the interaction of neural waves in the visual cortex, which was recognized as one of the computation mechanisms in the brain[40].

## Signal encoding and recognition

The sensitivity of the wave packet characteristics to the write position and the resulting high selectivity of the readout infer great efficiency of the proposed device as a neuromorphic reservoir, which processes the incoming signals that are encoded in the spatial-temporal distribution of the write laser intensity. We illustrate the reservoir function by recognizing "visual shapes" drawn by the write laser on the patterned surface. Symbols are drawn by sequential step-like changing of the relative position of the write laser spot along a selected trajectory formed from the set of the discrete positions as shown in Fig. 3. The resulting single waveform for a specific shape is integrated over the whole trajectory (Methods). For our system, we have arbitrarily selected the following six symbols for recognition: 'L', '−', 'O', '+', 'T' and 'Z'. The corresponding write laser trajectories and the idealized images of the symbols formed by the write laser spot are shown in Fig. 3b. Ten sets of symbols were drawn using this procedure. Figure 3c shows the exemplary readout waveforms.

In order to demonstrate the efficiency of our reservoir we minimize the number of characteristics used for recognition by the readout ANN. To attribute the information and statistical properties of the readout signals, we utilize only three characteristics: the Shannon entropy, $h$, the variance, $\sigma^2$, and the skewness, $\mu$ (see Methods). Figure 3d presents the distributions of the chosen parameters in the three-dimensional parameter space ($h$, $\sigma^2$, $\mu$). Each write laser trajectory is shown by a different colour. The plot reveals the formation of clouds corresponding to the specific symbols. The distribution of the parameters within each cloud is due to the uncertainty of the read laser spot position, which is of ±200 nm along all the axes and exceeds the spatial sensitivity of the readout (Fig. 2). As a result, each symbol shape is randomly distorted (see Supplementary Fig. 2 for details) and, thus, some clouds partially overlap (e.g. for the symbols 'T' and 'Z'). Nevertheless, the clear distinction of the parameters' clouds confirms the reservoir efficiency in filtering and sorting the encoded "visual shapes".

To produce a large set of randomly distorted "visual shapes" for the follow training of the ANN, we use the technique of data augmentation (mixing)[41]. We have assembled the corresponding "visual shapes" from randomly selected "pieces" (i.e., the waveforms of the respective discrete positions of the corresponding trajectories) from the ten pre-measured sets (Methods). 2000 readout waveforms for each symbol have been produced. The distributions of the chosen statistical parameters for the augmented symbols are shown in Supplementary Fig. 3. The randomized "visual shapes" form larger clouds with more significant overlap, but remain well separated. To exclude the artificial character of the parameters' distinction, we have checked that the waveforms, which do not correspond to the selected trajectories but consist of the same number of discrete positions, do not form separate clouds (see Supplementary Fig. 4).

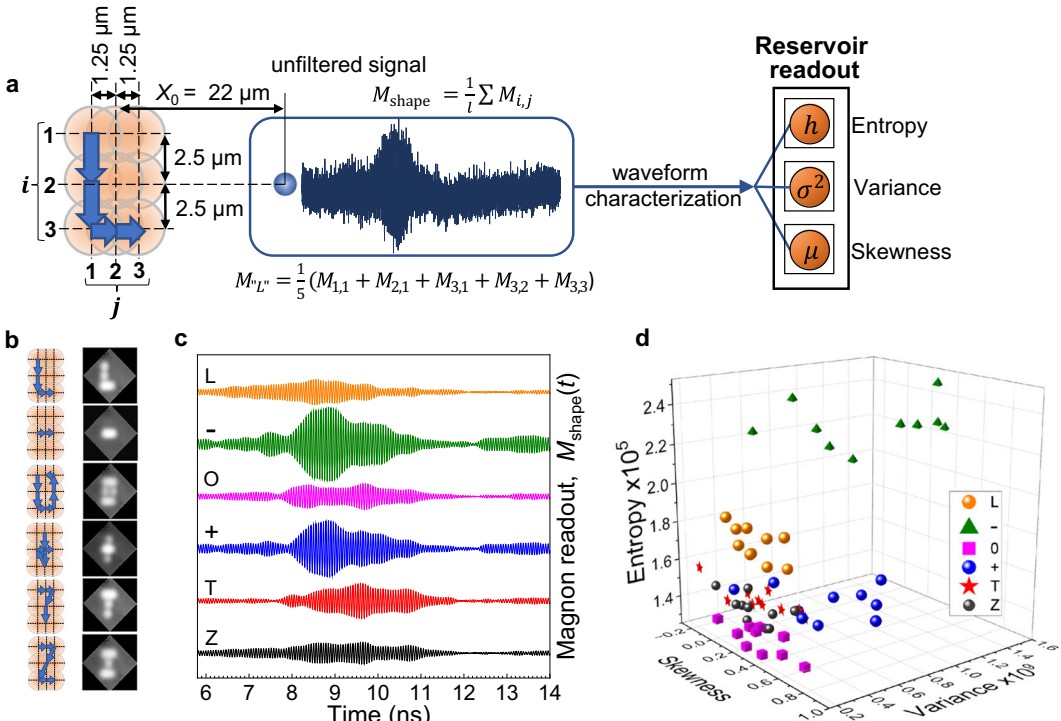

**Fig. 3 | Laser "drawing" of the "visual shapes" and their sorting. a** Encoding the visual shapes by sequential relative shift of the write and read spots and the following characterization of the integrated waveform. The waveform $M_{shape}(t)$ is the sum of the waveforms $M_{i,j}(t)$ measured in the corresponding discrete positions forming the selected trajectory normalized to their number, $l$. The trajectory illustrated on the left corresponds to the symbol 'L'. The respective exemplary waveform, $M_L(t)$,(as measured) is shown in the central frame. **b** Selected writing trajectories (left column) and the idealized artificially constructed images of the drawn symbols (right column). The micrographs are obtained by capturing images of the write laser spot on the nanograting surface sequentially shifted along the respective trajectory. The images with the spots forming the corresponding visual shape are overlapped in a graphic redactor. **c** Exemplary noise-free readout waveforms integrated over the respective trajectories. **d** Distribution of symbols in the parameter space spanned by the Shannon entropy, the skewness, and the variance, obtained from unfiltered magnon readouts for ten drawn trajectories for each symbol.

Since the reservoir demonstrates high selectivity in filtering symbols, that are well separated in the parameter space, various algorithms for the symbol classification can be realized. We have chosen the conventional physical reservoir approach[5,10] with training the readouts. To accelerate the training process, we use a simple ANN forming a feedforward Multi-Layered Perceptron with a back-propagation algorithm[42]. The readout ANN, which is schematically shown in the inset of Fig. 4a, consisted of only 11 nodes in three layers, and the single output layer. As output, the ANN returns an encoded integer to identify the letter present: 1 = 'L', 2 = '-', 3 = 'O', 4 = '+', 5 = 'T', and 6 = 'Z'. Before recognition, the ANN was trained with all six letters during, in total, 2000 epochs with a learning rate of 0.001. The ANN learning rate is the largest magnitude the parameters of the network can change within one epoch. It and the other ANN hyperparameters (the number of epochs, the early stopping patience, the random seed, the number of learning layers, and the number of nodes in each of those layers) were adjusted using a grid search approach to achieve the fastest training, but their absolute values can be varied without affecting the recognition. The evolution of the training and validation errors during training epochs are shown in Fig. 4a by blue and red colour, respectively. After ~400 epochs both training and validation errors saturate at values around 0.003, implying an accuracy of 99.7%. The recognitions of the individual symbols are summarized in the confusion matrix in Fig. 4b. The matrix shows that the symbols 'L','-', and 'O' are recognised with 100% accuracy. The symbol '+' is recognised in 84.5% cases, in 13.7% cases it is confused with the symbol 'T', and with a probability 1.8% is recognised as the symbol 'Z'. The symbol 'T' is recognized with a probability 83.6%, although in 4.2% of cases it could be confused with the symbol '+' and in 12.2% cases with the

symbol 'Z'. Finally, the symbol 'Z' is recognised with 86.1% accuracy, while it could be confused with the symbol 'T' with probability 13.6%, and in one case this symbol was unclassified (reflected as symbol 'Null' in Fig. 4b). The main reason for confusion relates to the similarity in coding of the symbols, see Methods, and could apparently be resolved by changing the coding protocol, i.e., the way in which the laser "writes" a symbol on the device surface. For example, if we exclude symbol 'Z' from the set of symbols, the ANN recognizes the remaining symbols with 100% accuracy (see Supplementary Fig. 5). The random waveforms, i.e. those not corresponding to the six selected "visual shapes", are randomly attributed to one of the shapes or not recognized.

To verify the key role of the phonon-magnon interaction in the reservoir functionality, we apply the above recognition procedure to the readout signals passed through different band-stop filters (see Supplemental Fig. 5). If the stopband of the filter was set to 15–20 GHz, which excludes the signal components within the frequency range corresponding to the first W-band, the ANN was unable to reliably distinguish the symbols from the set. The recognition becomes completely impossible if the stop band of the filter is 0–20 GHz, which also suppresses the contribution of bulk and surface phonon modes. However, cutting the frequencies above 20 GHz with preserving the main information-capable range (0–20 GHz) significantly reduces the recognition accuracy. It indicates the role of high-frequency components of the readout waveforms such as the high-order W-band harmonics, parametrically mixed first-order W-modes and broadband noise, which could play a constructive role by promoting mode mixing and enhance nonlinearity effects[43]. The separation and visualization of the specific high-frequency

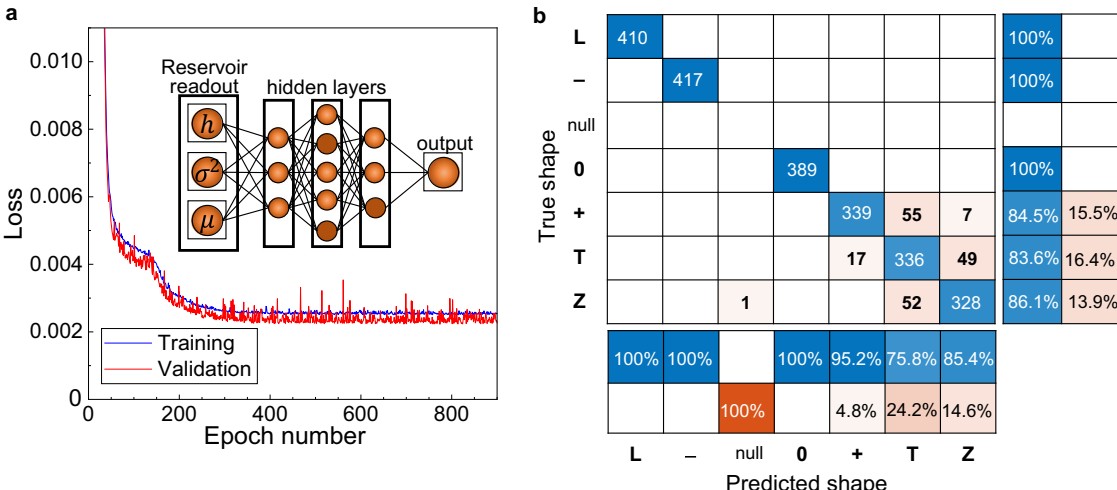

**Fig. 4 | Recognition of the "visual shapes". a** Dependence of the validation (red curve) and training (blue curve) errors on the number of epochs for the set of symbols. The inset shows a scheme of the readout ANN. **b** Confusion matrices for the set of symbols.

contributions and noise, however, are out of the scope of the present study.

## Discussion

### Robustness, miniaturization, and energy costs

Despite the simplified sequential encoding procedure, the reservoir demonstrates all the main properties required for efficient reservoir computing. The high sensitivity of the information and statistical characteristics of the generated readout to the relative position of the write and read laser spots allows us to realize RC without an input ANN. The high selectivity of the reservoir with respect to the superposition of phonon wavepackets generated at different 'write' laser positions allows recognition and classification of distorted and noisy "visual shapes". Despite unavoidable errors in the positions of the laser spots (up to 5% of the largest linear size of the symbol) and in the measurements performed on different days, the readout signals generated by the reservoir are reliably recognized by an ANN, which evidences its high robustness. The weak temperature dependences of the elastic and magnetic parameters of the used materials around room temperature[44–46] makes the reservoir functionality also resistant to the unavoidable temperature variations due to background heating.

The demonstrated reservoir functionality relies on the volatility of the phonon wavepacket provided by the interference of guided phonon modes. The working principle resembles the prototype concept with water waves in a bucket[11] but is realized at much higher carrier frequencies of ~10 GHz and on a drastically reduced spatial scale. The total area of the NG is $25 \times 100\,\mu m^2$, of which only $10 \times 30\,\mu m^2$ are used. The area used for the symbol encoding is $3.5 \times 6\,\mu m^2$, which is comparable in size with that of a single pixel in the CMOS-sensor of a modern digital camera. The readout spot diameter is below $1\,\mu m$ and the confirmed spatial resolution is ~100 nm. Such compactness is achieved in a nanodevice fabricated by state-of-the-art technology, which enables the implementation of the proposed reservoir on a chip. Moreover, the compactness is combined with large operational distances. In ambient conditions at room temperature, the information encoded into a multimode phonon wavepacket can be delivered on distances of tens of microns with protection from scattering at the surface and dephasing[28]. This facilitates a complex chip architecture and multithread reservoir computing with high information density.

The demonstrated magnon-phonon information processing also has very low energy costs. The phonon wavepackets are generated by the 'write' laser with 50 mW averaged power at 80 MHz repetition rate. The transmission of optical excitation to coherent phonons guided

along the surface has low energy efficiency, because a vast amount of the absorbed optical energy is converted to non-coherent phonons and bulk phonon modes escaping to the substrate. Each optical pulse with an energy of $6 \times 10^{-10}$ J is converted to a phonon wavepacket carrying only ~$10^{-16}$ J (see Methods for the details of estimations). This value is 5-orders of magnitude smaller than a single neural spike in the human brain, which estimated energy is ~10 pJ[8]. The averaged acoustic power involved into the signal processing is just about 10 nW. Due to these extremely low numbers, the signals are averaged over $10^{10}$ 'write' pulses/phonon wavepackets at every discrete point of the selected trajectory to achieve reasonable signal-to-noise ratio. It takes about 2 min for every step of the "visual shape" drawing and, thus, limits the operational speed. However, the total energy costs remain extremely low: the single symbol drawing costs about $5\,\mu J$ and the total acoustic energy, which would be spent for the drawing and processing of the symbols in the whole training process, is less than 5 mJ. For comparison, it is two-order of magnitude less than the acoustics energy transmitted by a RF acoustic filter during 1-s operation of a modern wireless communication device[47]. This estimated value does not include the energy costs of the reservoir peripherals, which, in our experiments, is based on non-optimized research laboratory equipment.

### Perspectives

The reservoir functionality in our experiment has been demonstrated by recognition of pseudo-visual information, but the proposed reservoir can process any type of signals converted to guided multimode phonon wavepackets. Thus, further development of the phonon-magnon reservoir should exploit an advancement of the write and read methods. One prospective direction here would be to implement the optical input as a two-dimensional, spatially distributed profile of laser intensity, enabling "instantaneous imprinting" of the information on the surface. This approach requires larger operational area than the sub-wavelength "drawing" used in our study but can be combined with parallel spatially distributed reading, which will improve the operational speed. Semiconductor mode-locked lasers whose efficiency in generation of coherent hypersonic phonon wavepackets has been recently proven[48] can provide the required miniaturisation to such an approach.

Another direction is real-time operation with changing information patterns. Nowadays reservoir computing is considered as one of the best machine learning frameworks for temporal or sequential data processing[49]. The ability of the presented device to operate in the GHz

frequency range promises high speed of neuromorphic operations. It requires, however, input methods with significantly higher energy efficiency. We consider a piezoelectric technique of generation of acoustic waves to be the most prospective in this context. Modern RF acoustic filters, which operate in billions of wireless communication devices, possess high (up to 90%) efficiency of electric to acoustic signal conversion[47]. The carrier frequency is currently limited by several GHz but has to be extended due to the requirements of the next (6 G) communication standards. Prototype devices already demonstrate an energy conversion efficiency of several per cent at frequencies above 10 GHz[50,51]. It will allow real-time operations using the proposed reservoir architecture keeping the total energy costs very low. Moreover, a two-order of magnitude increase of the strain amplitudes will drastically enhance the contribution of magnon-phonon nonlinearity[39], which is manifested only indirectly in our study.

Recent proposals indicate that wave phenomena in the brain could constitute alternative information processing, complementary to the commonly accepted mechanism of learning via developing connections between neurons[52]. For example, it has been shown that the brain can achieve selectivity function as well as stimulus detection via interference of neural waves, even if the tissue is fixed and does not accumulate information via learning[40]. In this case, the brain tissue operates as wave reservoir supporting the interaction of waves with two components—excitatory and inhibitory, reminiscent of the interacting phonons and magnons in our reservoir. With this mechanism, information processing in the visual cortex is realised via excitatory-inhibitory wave interference or mixing, even further deepening the analogy with the proposed on-chip phonon-magnon reservoir. More generally, cognitive functions could be understood in terms of spatio-temporal pattern formation[53]. Therefore, the role of standing and travelling waves in brain activities attracts significant attention[54]. The understanding of neuromorphic functions in terms of waves interaction become increasingly more important with development of quantum AI and quantum reservoir computing[55–58], where waves phenomena play the prime role. All this indicates potential and possibility for a new type of neuromorphic technology—wavetronics— where elements utilise travelling and standing waves for realisation of neuromorphic functions. Our results contribute to the development of this novel technology.

## Summary

We have designed and fabricated a hybrid semiconductor–ferromagnetic nanodevice for reservoir computing. The reservoir has optical input and output and encodes the input signals into multimode phonon wavepackets propagating in a semiconductor phonon waveguide. The neuromorphic function is realised through the interaction of the multimode input-sensitive phonon wavepackets with multiple magnon modes, which provides a high separability of the reservoir. The efficiency of the reservoir is demonstrated by reliable recognition of symbols "drawn" by the laser on the nanodevice surface in an area of several square microns, which is comparable with the size of the write laser spot. Our work paves the way for developing novel ultra-compact wave-based neuromorphic architectures on chips.

## Methods

### Device production

The examined nanodevice was epitaxially grown on a commercial GaAs substrate [(100)-semi-insulating GaAs]. First, the phonon waveguide consisting of 10 pairs of AlAs/GaAs layers of 51-nm and 72-nm thickness, respectively, was fabricated by molecular beam epitaxy. Then, the ferromagnetic $Fe_{0.81}Ga_{0.19}$ layer was deposited by magnetron sputtering. The nanogratings were fabricated through focused ion beam milling (Raith VELION) with 100-nm wide grooves and ridges over an area of 25 μm × 100 μm. To ensure high beam resolution during the patterning of the gratings a low $Ga^+$ beam current of 22 pA was applied at 35 keV beam energy. The milling dose was set to 0.3 nC/μm², resulting in the groove depth of 25 nm.

### Optical writing and reading

The scheme of signal encoding and read-out exploits two mode-locked Erbium-doped ring fibre laser oscillators (TOPTICA FemtoFiber Ultra 1050 and FemtoFiber Ultra 780). The lasers generate pulses of 150-fs duration with a repetition rate of 80 MHz at wavelengths of 1050 nm (write) and 780 nm (read). The write beam was focused by a microscope objective (×20 magnification; N.A. = 0.4) to the backside of the ferromagnetic layer, through the GaAs substrate and GaAs/AlAs phonon waveguide, which are transparent at the wavelength of 1050 nm. The focused pump spot had a Gaussian intensity distribution $\sim\exp\left(-\frac{r^2}{2R^2}\right)$, where $r$ is the distance from the centre of the spot and $R = 1$ μm is the spot radius at the $1/\sqrt{e}$ level. The maximal used pulsed excitation density was 2 mJ/cm². The linearly polarized readout beam was focused by another microscope objective (100× magnification; N.A. = 0.8) to the front side of the nanodevice. The focused readout spot also had a Gaussian distribution with $R = 0.3$ μm and an energy density of 1 mJ/cm². Measurement of the read-out signals was based on the polar magneto-optical Kerr effect, i.e., the rotation of the readout pulse polarization plane by an angle proportional to the normal ($z$-axis) projection of the net magnetization. The oscillations in the Kerr rotation signal reflect the precession of the magnetization[59], which is a coherent superposition of the magnon modes driven by the phonon wavepacket[60]. The polarization rotation was detected in a differential scheme by a balanced photoreceiver with a 10-MHz bandwidth. Time resolution was achieved employing an asynchronous optical sampling (ASOPS) technique[61]. The write and read oscillators were locked with a frequency offset of 1600 Hz, which changed gradually the delay between the write and read pulses. In combination with the 80-MHz repetition rate and 10-MHz photoreceiver bandwidth, it allows measuring time-resolved signals in a time window of 12.5 ns with a time resolution of ~1 ps. The relative position of the write and read spots is controlled by two independent piezo-translators with a precision of 0.05 μm. The distance $X_0 \geq 20$ μm between the write and read spots was chosen to minimize the contribution of the first-order surface phonon modes, which are intensively scattered by the patterned surface[28]. For visualization of the laser spots and control of their relative positions and focusing, a magnified (×75) image of the nanodevice surface was focused on crossed micrometre slits mounted at the intermediate focal plane between the nanodevice and a photoreceiver, where it was captured by a microscope (×6.5 magnification) with CMOS camera. To show more explicitly the volatility of the readout waveform, the readout signals presented in Figs. 2d, f, g, and 3c were passed through a 15–20 GHz band pass filter, which eliminates the broadband noise.

### Sequential drawing of the visual shapes

We used sequential "drawing" of visual shapes by step-by-step change of the relative position of the write and read laser spots in accordance with the selected trajectory as illustrated in Fig. 3. In the $XY$ coordinate frame centred at the apex of the readout spot as shown in Fig. 2, the relative position of the write spot form a 3×3 coordinate matrix as shown in Fig. 3a. The positioning of the write and read spots for each point of the selected trajectory was automatized with the following algorithm: (i) randomized positioning of the read spot on the device surface (ii) adjusting the focusing of the read beam, (iii) adjusting the focusing of the write beam and the spatial overlap of the write and read spots, (iv) shift of the read spot to the required position. The experimentally estimated error of the automized positioning was 0.2 μm for all three-coordinate axis. The signal in every point of every trajectory was averaged over 10,000 measurements. Each trajectory was measured 20 times with averaging of the resulting waveform. 10 sets with six selected trajectories in each were measured.

## Drawing of randomized visual shapes

For augmentation of the selected visual shapes with randomized distortion for the following recognition we measured 10 sets with nine signals in each set. Each of the nine signals corresponds to the specific relative positions of the centres of the write and read spots as described above. The algorithm of the automated positioning of the write and read spots was the same as for the sequential drawing with the same positioning errors. The waveform for each point was averaged over 200,000 measurements. The 10 sets were measured on three experimental days (4 sets in the first day, and 3 sets each on the second and third days) with a complete restart of the experimental setup. The waveforms for recognition were obtained by random summation of the corresponding signals from all 10 sets. The recognized shapes were obtained by the following normalized summations:

$$
\begin{aligned}
"L" &= (M_{1,1} + M_{2,1} + M_{3,1} + M_{3,2} + M_{3,3})/5; \\
"-" &= (M_{2,1} + M_{2,2} + M_{2,3})/3 \\
"O" &= (M_{1,1} + M_{1,2} + M_{1,3} + M_{2,1} + M_{2,3} + M_{3,1} + M_{3,2} + M_{3,3})/8; \\
"+" &= (M_{1,2} + M_{2,1} + M_{2,2} + M_{2,3} + M_{3,2})/5; \\
"T" &= (M_{1,1} + M_{1,2} + M_{1,3} + M_{2,2} + M_{3,2})/5; \\
"Z" &= (M_{1,1} + M_{1,2} + M_{1,3} + M_{2,2} + M_{3,1} + M_{3,2} + M_{3,3})/7.
\end{aligned}
\tag{1}
$$

2000 waveforms for each symbol were randomly generated (the randomized set for "-", which consists of only 1000 unique waveforms, included 1000 additional randomly repeated waveforms). Only the integrated waveforms were processed. The separate waveforms $M_{i,j}(t)$ measured in the corresponding positions of the write laser beam were neither processed nor used to support the following recognition.

## Waveform characterization

For characterisation of the readout signals we use the following quantities: the Shannon entropy, $h$, the variance, $\sigma^2$, and the skewness, $\mu$. The signals were processed as measured, without filtering and subtracting the dc-background of the photoreceiver. To calculate $h$, we split the interval between the minimal and maximal voltage of the readout signals ($-4.02$ mV and $-3.05$ mV, respectively) into $J = 10^5$ bins, and then apply the formula

$$
h = -\sum_{j=1}^{J} p_j \ln p_j,
\tag{2}
$$

where $p_j$ is the probability that the readout voltage has a value within the $j^{\text{th}}$ bin. For calculating the statistical characteristics, we utilised the formulas below:

$$
\sigma^2 = \frac{1}{N} \sum_{i=1}^{N} (V_i - \bar{V})^2,
\tag{3}
$$

$$
\bar{V} = \frac{1}{N} \sum_{i=1}^{N} V_i,
\tag{4}
$$

$$
\mu = \frac{1}{N\sigma^3} \sum_{i=1}^{N} (V_i - \bar{V})^3,
\tag{5}
$$

where $V_i$ is the value of the discretised in time readout voltage at the $i^{\text{th}}$ time step, and $N$ is the total number of time steps.

## Artificial neural network

For processing the properties of the signals to classify the letters from the device, we employ a simple Artificial Neural Network (ANN), illustrated in the inset of Fig. 4a. The network is made of the input layer (where the properties are fed into the network) with 3 nodes (one for each property of the signal), then we have the first learning layer made up of 3 nodes, the second learning layer with 5 nodes, and the final learning layer of 3 nodes, with one final output layer. The output layer gives the classification of the input signal. Each of our layers is connected with the Rectified Linear Unit (ReLU) activation function, apart from the final learning layer connected to the output layer, which is a linear activation function. The network is coded within the Python module PyTorch[62]. We use the Adam optimiser[63] to train the network, with a Root Mean Squared Error (RMSE) loss. The network optimises its weights and biases by backpropagating the error through its connections and adjusting the values of the network's parameters. The adjustments are made on the order of the learning rate, here taken to be 0.001. We allow the network to train for 2000 epochs. However, employing an early stopping procedure, we stop the network from overfitting the training dataset. This is further achieved by splitting the dataset of letters into non-overlapping training, validating, and testing sets, with a standard 60, 20, 20% split[64].

## Numerical modelling

The S- and W-modes' profiles and the excitation and propagation of phonon wavepackets were modelled using COMSOL Multiphysics® software. The spatial profiles and dispersions of the S-modes were obtained by solution of the Landau-Lifshitz and Maxwell equations in the frequency domain[36]. The following parameters of $Fe_{0.81}Ga_{0.19}$ were used for the calculations: saturation magnetization, $\mu_0 M_s = 1.76$ T, exchange stiffness, $D = 210^{-17}$ T m$^2$, cubic and uniaxial anisotropy coefficients of 23 mT and 11 mT, respectively. For the calculation of the W-modes spatial profiles and spatial-temporal evolution[28], we used the following stiffness tensor components, $C_{kl}$, and the mass density, $\rho$: for $Fe_{0.81}Ga_{0.19}$ $C_{11} = 209$ GPa, $C_{12} = 156$ GPa, $C_{44} = 122$ GPa, $\rho = 7800$ kg/m$^3$; for GaAs $C_{11} = 119$ GPa, $C_{12} = 53.8$ GPa, $C_{44} = 59.5$ GPa, $\rho = 5316$ kg/m$^3$; for AlAs $C_{11} = 119.9$ GPa, $C_{12} = 57.5$ GPa, $C_{44} = 56.6$ GPa, $\rho = 3760$ kg/m$^3$. We used a model nanograting structure with 150 periods (i.e. 30 μm long in $x$ direction). The structure was limited in $z$ direction by the size of 1.9 μm (500 nm below the phonon waveguide), and the low-reflecting boundary condition was applied to the bottom boundary to minimize unwanted reflections. Free boundary conditions were applied to the top and side boundaries of the structure. The structure was considered uniform in $y$ direction. The excitation of the structure by the optical pulse creates thermal stress which was simulated using a two-temperature model for the lattice temperatures of the ferromagnetic cap and phonon waveguide[65]. Then, the calculated thermal stress was used as the input data for the acoustic modelling. The calculations were carried out in the time domain up to 6 ns. The results of the simulation are presented in the Supplemental Video and in Fig. 2. The energy of a single W-mode wavepacket was estimated as $U = \frac{1}{2}\upsilon\langle c_{11}\rangle\langle\eta^2\rangle \approx 5 \cdot 10^{-17}$ J, where $\upsilon = \pi R^2 h$ is the total volume of the wavepacket ($R = 1$ μm is the write laser spot radius and $h = 1.3$ μm is the phonon waveguide thickness, $\langle c_{11}\rangle = 119.5$ GPa is the averaged stiffness constant of the GaAs/AlAs waveguide and $\langle\eta^2\rangle \approx 2 \times 10^{-10}$ is the averaged square of the strain amplitude.

## Data availability

The raw and processed experimental data generated in this study have been deposited in the Zenodo database under https://doi.org/10.5281/zenodo.8411702.

## Code availability

The code used for the ANN training is available in the Zenodo database under https://doi.org/10.5281/zenodo.10080655.

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

## Acknowledgements

We are grateful to Andrey Akimov for fruitful discussions. A.V.S. and M.B. acknowledge the support by the Deutsche Forschungsgemeinschaft through the project SFB TRR 160 (project A1). The cooperation between TU Dortmund and the Lashkaryov Institute was supported by the Volkswagen Foundation (Grant 97758). M.B. acknowledges the support by the research centre "Future Energy Materials and Systems" of the Research Alliance Ruhr.

## Author contributions

D.D.Y. constructed the experimental setup, carried out the experiment, and performed signal characterization and analysis, A.V.S. designed the reservoir device and experiment and supervised the experiment and theoretical modelling. L.D.C. designed and coded the artificial neural network (ANN), processed and analysed signals, and performed training and testing of the ANN, S.M.K. designed the reservoir device and performed theoretical analysis and numerical modelling. A.N., R.C. and A.W.R. designed and produced the reservoir device, S.S. and A.G.B. supervised signal processing and the ANN's design, coding and training, A.V.S., S.S. and A.G.B. conceptualized the manuscript, D.D.Y., A.V.S., L.D.C., S.M.K, A.W.R., S.S., A.G.B., and M.B. discussed the results and wrote the manuscript.

## Funding

## Competing interests

The authors declare no competing interests.
