## [Peer Review File · Nature Communications]

REVIEWER COMMENTS

Reviewer #1 (Remarks to the Author):

The authors describe a novel material in a waveguide structure that allows the guiding of photons and magnons (polarons). Many polarons are excited by an ultrafast laser focused to a small spot size, which excites different mode amplitudes depending on the position of the focused laser. At a separate location, a second laser reads out the polarons. The authors use this system as a nonlinear substrate for realizing a so-called reservoir computer, which is a recurrent artificial neural network. They use the system to identify some symbols "drawn" on a 3x3 grid of spots. The drawing happens sequentially - they write one spot in the symbol, then reposition the laser and write the next one, etc. For each spot, they readout the polaron signal and then process the data to classify the symbol.

The authors use two primary motivations for the work. I find that their results do not support either motivation. Once these motivations are removed, this is really a highly specialized paper on the optical excitation of the material. I do not think that the work is suitable for Nat Comm and I recommend that it be rejected. The authors might consider submitting the work to a more specialized journal.

Motivation on the "ultimate size" device.

The authors claim that their approach will eventually produce the smallest reservoir computer, operating with a small size and power. Their presentation does not support this claim. First, they do not consider the fact that they need an ultrafast laser for writing the symbols and reading out the polarons. Second, they need a high-speed device to record the signal. Third, they have to perform substantial processing of the signal to find the entropy, etc. Fourth, they need a traditional artificial neural network running on a standard computer to process the data.

Based on this last point, it is not at all clear that there is actually any advantage of using polarons given the very high resource count for both the input and output stages of the system.

The motivation for a "visual" processing system

I also find that this motivation is not supported by the presented work. Their device appears to require coherent light from an ultrafast laser. The device also has a narrow spectral resonance. Based on these facts, it appears that there is no way that this system can be used for natural light similar to what is sensed by the eye.

Other comments:

- The sequential writing of the patterns is not acceptable for pattern recognition. There will be interactions of the polarons when writing all spots simultaneously vs. one at a time as done in the current experiment. In this sense, they are only classifying spots on a 3x3 grid, not the symbols as they claim.

Reviewer #2 (Remarks to the Author):

Manuscript # NCOMMS-23-17402-T

Title On-chip phonon-magnon reservoir for neuromorphic computing

The paper suggests a novel promising approach in building an on-chip reservoir for neuromorphic computing, which addresses the long-standing problems of miniaturization, efficiency, and robustness – main stumbling blocks in the practical realization of neuromorphic computers. The work proposes an elegant design of a miniature “liquid-state machine”, where the reservoir function is realized through the interaction of phonon and magnon-modes. The reservoir consists of semiconductor heterostructures, accommodating multimode phonon waveguides, capped by a nanograting patterned in a metal ferromagnetic layer hosting magnons.

This approach offers a new realization of unconventional reservoir hardware using waves on nano-chip, which is the next potential step of neuromorphic technology. Such wave-tronics approach is a complimentary to a standard realization of reservoirs by artificial neural networks with fixed connections. The authors convincingly demonstrate the high efficiency of such reservoir in the recognition of optically input symbols.

I believe that the presented results are solid and an important step in the development of practical neuromorphic computing. It can be considered for publications after the authors addresses the following issues in the actual manuscript, not just the reply to the referees.

1) the idea of wave-tronic paradigm is promising for future development neuromorphic technology. However, the authors should better explain the essence of the concept, especially in the content of the role of waves in the functioning of the biological cortex, a research area which is now booming in neuroscience.

2) The author use point-by-point approach to input symbols, which obviously not the best way to input the reservoir because it does not promote wave mixing. Does this imply that their reservoir is linear. Can the authors comment if/how the nonlinearity is realized in their reservoir.

3) Author state that the readout ANN was trained with learning rate with 0.001. Does the result depend on the learning rate?

4) It is known that noise plays an important role in reservoir computing, but this discussion is missed in the paper. Could the authors to explain this better?

5) Additional relevant papers could be added in the reference list, to make the bibliography more complete.

All of the explanations added must be in the actual manuscript, not just in the reply to the referees, which is not available to the readers.

Reviewer #3 (Remarks to the Author):

In this manuscript the authors propose a neuromorphic computing device based on the phonon-magnon interaction in a phonon wave-guide patterned with a ferromagnetic layer acting as the reservoir for computing. For that goal, a pulsed laser (input signal) is converted into a propagating

phonon wave packet that travels along the wave-guide and interact with the magnon modes, leading to polar magneto-optical Kerr effect that is detected (output signal) at a distant location.

The topic of neuromorphic computing and the exploration of different hardware platforms is currently of great interest. In this sense the present work might represent an interesting step forward in the use of phonons and magnons as suitable candidates for the reservoir.

There are some points, however, that in my opinion need to be further addressed or clarified. I now list them:

a) In the introduction section (line 63) the word 'quanta' is used to refer to photons, phonons and magnons used in previous implementations using different physical waves. To me, this gives the impression that something intrinsically 'quantum' is being implemented while in fact, at least in the present proposal, the physical waves are in a 'classical' regime, being mostly its wave nature the one exploited. I think it might deserved some clarification in the current context with many quantum computing proposals.

b) The authors mention that 'the laser pulses are converted in the device to a coherent multimode phonon wave-packet' but never explain the mechanism behind that conversion. Given that the coherence of the phonon is relevant, I suggest a brief explanation to be added.

c) Since there is a strong sensitivity of $M(t)$ to the characteristics of the W phonon modes, in particular initial amplitude and phase, is it to be expected that temperature changes or sample to sample variations could lead to relevant changes on the calibration of the device?

d) I do not understand the signals shown in the right panel of fig 3b. Why are they so clean and not noisy as the ones shown in fig3a or in supplementary fig 2?

e) It is my understanding that in order to calibrate the device, the writing laser is positioned on each one of the 3×3 pixels, the corresponding M_{ij} signal recorder. These signals are then superimposed according to the pattern and then the entropy, variance and skewness of it calculated.

What it is a bit unclear to me is the testing procedure: are the corresponding M_{ij} signals of the intended (unknown) pattern to be recognized all add up and then the 3 parameters calculated? Or are the different combinations that defined the 'recognized shaped' calculated to see which one match better?

f) What happens if the pattern that is intended to be read is not one of the 'recognized patterns', say a X?

g) What about the time scale for writing? Can one use a time dependent pattern for writing? What are the limitations for that? The author speculate on that in the conclusion section, please elaborate.

We are grateful to the reviewers for their careful examination of our manuscript, considerate criticism, and insightful questions. To address all critical comments and answer several specific questions we have performed additional investigations, namely:

- We have extended the set of “visual shapes” for recognition by waveforms integrated over the corresponding laser trajectories. These “directly-drawn” “visual shapes” form well separated clouds in the parameters’ space, confirming the reservoir functionality (Figure 3).
- We have modelled the spatial profiles of the interacting phonon (W-) and magnon (S-) modes in order to emphasize the spatial mode mixing in the resulting waveform.
- We have also added fast Fourier transform spectra of the readout waveforms measured at various relative positions of the write and read laser spots (Figure 2). The modification of the FFT spectra induced by the relative shift of the write and read laser spots on distances smaller than the phonon wavelengths illustrate the mode mixing in the spectral domain.
- To illustrate the nonlinear character of the suggested reservoir, we have extended the analysis of the spectral dependence of the recognition accuracy.
- To confirm the high application potential of the suggested approach, we have estimated the acoustic energy costs of all the operations performed by the reservoir in our experiment and suggested a more specific scenario of a following application-focused development.

Below we give detailed answers to all reviewer remarks along with the corresponding modifications made in the manuscript. The line numbers and references correspond to the revised version of the manuscript.

Reviewer #1.

The authors describe a novel material in a waveguide structure that allows the guiding of photons and magnons (polarons). Many polarons are excited by an ultrafast laser focused to a small spot size, which excites different mode amplitudes depending on the position of the focused laser. At a separate location, a second laser reads out the polarons. The authors use this system as a nonlinear substrate for realizing a so-called reservoir computer, which is a recurrent artificial neural network. They use the system to identify some symbols "drawn" on a 3x3 grid of spots. The drawing happens sequentially - they write one spot in the symbol, then reposition the laser and write the next one, etc. For each spot, they readout the polaron signal and then process the data to classify the symbol.

The authors use two primary motivations for the work. I find that their results do not support either motivation. Once these motivations are removed, this is really a highly specialized paper on the optical excitation of the material. I do not think that the work is suitable for Nat Comm and I recommend that it be rejected. The authors might consider submitting the work to a more specialized journal.

Answer

We are grateful to Reviewer 1 for carefully examining our work and the expressed criticism, but we respectfully disagree with the main statement of the reviewer, as well as the specific comments. To meet the criticism, we have performed additional measurements and numerical modelling to support our concept, and amended the text. We hope that our response below and the corresponding modifications made in the manuscript will better transform the message of the paper and justify our conclusions.

Reviewer #1. Remark 1

Motivation on the "ultimate size" device.

The authors claim that their approach will eventually produce the smallest reservoir computer, operating with a small size and power. Their presentation does not support this claim. First, they do not consider the fact that they need an ultrafast laser for writing the symbols and reading out

the polarons. Second, they need a high-speed device to record the signal. Third, they have to perform substantial processing of the signal to find the entropy, etc. Fourth, they need a traditional artificial neural network running on a standard computer to process the data.

Based on this last point, it is not at all clear that there is actually any advantage of using polarons given the very high resource count for both the input and output stages of the system.

Answer

We are sorry if the original manuscript produced the impression that our aim was to develop “the smallest reservoir computer”. We did not intend to make such a bold claim.

However, our device is indeed compact, and the size of the device is of key importance for advancing RC technologies. A hardware reservoir of comparable size ($\sim 100 \mu\text{m}^2$) has been demonstrated only in several recent spin-wave-based concepts [23,24]. The device suggested in our work, consists only of a substrate, a GaAs/AlAs phonon waveguide, and a patterned ferromagnetic nanolayer. It operates according to the physical reservoir computing framework (see, for example, Ref. [10]).

The demonstrated compactness is achieved by exploiting acoustic phonons of hypersound frequencies as information carrier. Information transmission by acoustic waves is the key technology in modern wireless communications. Because the frequency range used in our study targets the next 6G communication standard, we believe that quick adaptation of our invention for real-life applications is surely possible.

Note that the lasers and readout components are not parts of the reservoir. We do not optimize these parts of the system, which is subject to future investigations. In fact, the input and output could be realized differently, e.g. using piezo-electric transducers, or even by applying electric or magnetic fields. Nevertheless, the standard laboratory equipment mentioned by the reviewer as a “high-speed device” is not a bottleneck for potential applications. It serves as periphery of the reservoir computing system in the same manner as in many other neuromorphic physical concepts (see, for example, Ref [27]) and includes a photoreceiver with a bandwidth of 10 MHz and a digital oscilloscope operated at the sampling rate of 20 MS/s and triggered at 1.6 kHz (see Methods). These low rates enable real-time calculation of the signal parameters by a pocket electronic device, e.g. a smartphone.

To explain in detail why we are confident with the high application potential of our device, despite the “high resource count” of its first experimental realization, we have significantly expanded the discussion part of the manuscript. Now, it includes the results of numerical modelling which estimates the energy costs of the performed operations and a more detailed description of the application perspectives.

Lines 257 -271

The demonstrated magnon-phonon information processing has also very low energy costs. The phonon wavepackets are generated by the ‘write’ laser with 50 mW averaged power at 80 MHz repetition rate. The transmission of optical excitation to coherent phonons guided along the surface has low energy efficiency, because a vast amount of the absorbed optical energy is converted to non-coherent phonons and bulk phonon modes escaping to the substrate. Each optical pulse with an energy of 6×10^{-10} J is converted to a phonon wavepacket carrying only $\sim 10^{-16}$ J (see Methods for the details of estimations). This value is 5-orders of magnitude smaller than a single neural spike in the human brain, which estimated energy is ~ 10 pJ [8]. The averaged acoustic power involved into the signal processing is just about 10 nW. Due to these extremely low numbers, the signals are averaged over 10^{10} ‘write’ pulses/phonon wavepackets at every discrete point of the selected trajectory to achieve reasonable signal-to-noise ratio. It takes about 2 minutes for every step of the “visual shape” drawing and, thus, limits the operational speed. However, the total energy costs remain extremely low: the single symbol drawing costs about 5 μ J and the total acoustic energy spent for the whole training process is less than 5 mJ. This value does not include the energy costs of the reservoir peripherals, which, in our experiments, is based on non-optimized research laboratory equipment.

Lines 284 - 296

Another direction is real-time operation with changing information patterns. Nowadays reservoir computing is considered as one of the best machine learning frameworks for temporal or sequential data processing [48]. The ability of the presented device to operate in the GHz frequency range

promises high speed of neuromorphic operations. It requires, however, input methods with significantly higher energy efficiency. We consider a piezoelectric technique of generation of acoustic waves to be the most prospective in this context. Modern RF acoustic filters, which operate in billions of wireless communication devices, possess high (up to 90%) efficiency of electric to acoustic signal conversion [49]. The carrier frequency is currently limited by several GHz, but has to be extended due to the requirements of the next (6G) communication standards. Prototype devices already demonstrate an energy conversion efficiency of several per cent at frequencies above 10 GHz [50,51]. It will allow real-time operations using the proposed reservoir architecture keeping the total energy costs very low. Moreover, a two-order of magnitude increase of the strain amplitudes will drastically enhance the contribution of magnon-phonon nonlinearity [39], which is manifested only indirectly in our study.

Reviewer #1. Remark 2

The motivation for a "visual" processing system

I also find that this motivation is not supported by the presented work. Their device appears to require coherent light from an ultrafast laser. The device also has a narrow spectral resonance.

Based on these facts, it appears that there is no way that this system can be used for natural light similar to what is sensed by the eye.

Answer

1) We did not aim to construct a visual processing system in our work, and there is no such claim in the manuscript, although in part our research was indeed inspired by the human visual system. Since the signals in our device are generated optically, we find the pseudo-visual representation of the encoded information for recognition to be the most explicit way to demonstrate the device's functionality. We have modified accordingly the conceptualized description of the reservoir operations and made respective changes in Fig.1:

Lines 90 – 98

The output waveform is integrated over the drawn trajectory corresponding to a specific "shape" for the following recognition by an ANN programmed in a personal computer. We may compare this procedure with handwriting and conceptualize the reservoir operations as recognition of a handwritten character. Thus, in analogy with the human vision (Fig. 1d), we consider the patterned surface as the retina, the phonons and magnons play the role of photoreceptors, the optical readout mimicking the optical nerve transferring the readout signal to the brain, namely ANN. The sub- μm wavelength of phonons and magnons allows achieving a large number of "receptors" per area ("receptor" density) and make the readout sensitive to a tiny change of the optical input. It enables recognition of the symbols drawn on the scale of just several wavelengths of the 'write' laser.

2) The laser pulse is transformed to the phonon wavepacket through the optically induced heating of the metal (Fe,Ga) film and is not limited to a specific optical spectral band. A detailed explanation of the phonon generation process has been added to the manuscript:

Lines 100 – 106

The physical processes involved in the operation of the fabricated nanodevice are demonstrated in Fig. 2. The 100-fs pulse of the write laser, focused on the metallic ferromagnetic layer to a spot of 3- μm diameter instantaneously induces thermal stress leading to the generation of the coherent acoustic wavepacket of hypersound (up to ~ 100 GHz) frequencies [30,31]. Due to the periodic surface patterning and the corresponding spatial modulation of the optically-induced stress along the x -axis, coherent phonons acquire the wavevectors: $k_x = n 2\pi/d$, where $d = 200$ nm is the nanograting period and $n = 1,2,3\dots$ is the harmonic order [32].

3) Any information delivered in any form must be converted to a specific signal for the following recognition. This is also true for human vision and hearing: the information supplied by light or sound is converted to neural (electric) signals transmitted for recognition to the brain. In our device, the input data is encoded in the phonon wavepackets. Visual information delivered by natural light can be converted to an acoustic signal as it is routinely done for photographs or videos wirelessly transmitted every day: CMOS sensor – RF Signal – Acoustic signal – A specific realization for a specific information type is an engineering task beyond the research presented here. We have added the corresponding remark to the discussion (see the previous reply, **Lines 284-296**).

Reviewer #1, Remark 3

- The sequential writing of the patterns is not acceptable for pattern recognition. There will be interactions of the polarons when writing all spots simultaneously vs. one at a time as done in the current experiment. In this sense, they are only classifying spots on a 3x3 grid, not the symbols as they claim.

Answer

We respectfully disagree with these statements.

1) Sequentially forming the signal for the following recognition is very common in everyday life. For example, handwriting is a sequential process forming a visually recognizable shape. Speech is another example. Recognition of sequentially coded information also has a significant advantage of prediction, e.g., the prediction of the trajectory of a moving object. We do not see why it is not suitable in the context of our work.

2) In our experiment, the resulting signal for a specific shape is a single waveform. It is integrated over the trajectory of the laser spot (directly or simulated) on the reservoir surface. We neither process the signals from the discrete write laser positions nor deconvolute the resulting signal. We have added the following remark to Methods section:

Lines 390-391

The separate waveforms $M_{i,j}(t)$ measured in the corresponding positions of the write laser beam were neither processed nor used to support the following recognition.

3) Each waveform (either from a single spot or integrated over a specific trajectory) is already a phase-sensitive superposition of many modes linearly and nonlinearly mixed by magnon-phonon interaction. This was not properly described in the original manuscript. To emphasize this fact we have extended Figure 2 with the W-modes' individual profiles and the fast Fourier transform spectra of the readout waveforms. We also extended the description of the role of magnon-phonon interaction (see also the reply to Remark 2 of Reviewer 2):

Lines 126 – 136

For any pair of phonon and magnon modes, the efficiency of their interaction depends on their spatial and spectral matching [37,38]. The spatial matching is determined by the mode profiles set by the device structural design. Figure 2b shows the calculated spatial profiles of the 22 first-order W-modes and the six lowest S- modes in the ferromagnetic layer. Their interaction occurs through the uniaxial (xx) and shear (xz) strain components of the W-modes [37] and for any pair of the illustrated W- and S-modes the overlap integral is nonzero. The spectral matching is set by the external magnetic field. The highest interaction efficiency takes place at the resonance condition when the frequencies of the interacting modes are equal. However, due to the finite spectral widths of the S- and W- modes, their interaction is still quite efficient when the modes are detuned. Moreover, magnon-phonon interaction is intrinsically nonlinear and is accompanied by the parametric effects, which result in frequency mixing of the phonon modes [39].

4) Discrete sequential coding is used due to the lengthy acquisition of the readout signals. It also allows us to simulate randomized shapes for training and recognition by the ANN instead of drawing thousands of "visual shapes". This process known as data augmentation is a routine procedure in machine learning (see Ref [41]).

We agree that the original manuscript did not adequately describe the encoding procedure and reservoir operations. Thus, we have made additional measurements (direct sequential drawing of the visual shapes) described in the revised Methods section and modified the corresponding part of the text and Figure 3.

Lines 168 – 174

Symbols are drawn by sequential step-like changing of the relative position of the write laser spot along a selected trajectory formed from the set of the discrete positions as shown in Fig. 3. The resulting single waveform for a specific shape is integrated over the whole trajectory (Methods). For our system, we have arbitrarily selected the following six symbols for recognition: 'L', '-', 'O', '+', 'T' and 'Z'. The corresponding write laser trajectories and the idealized images of the symbols formed by the write laser

spot are shown in Fig. 3b. Ten sets of symbols were drawn using this procedure. Figure 3c shows the exemplary readout waveforms.

Lines 197 – 195

To produce a large set of randomly distorted “visual shapes” for the follow training of the ANN, we use the technique of data augmentation (mixing) [41]. We have assembled the corresponding “visual shapes” from randomly selected “pieces” (i.e., the waveforms of the respective discrete positions of the corresponding trajectories) from the ten pre-measured sets (Methods). 2000 readout waveforms for each symbol have been produced. The distributions of the chosen statistical parameters for the augmented symbols are shown in Supplementary Fig. 3. The randomized “visual shapes” form larger clouds with more significant overlap, but remain well separated. To exclude the artificial character of the parameters’ distinction, we have checked that the waveforms, which do not correspond to the selected trajectories but consist of the same number of discrete positions, do not form separate clouds.

Lines 365 – 378

Sequential drawing of the visual shapes

We used sequential “drawing” of visual shapes by step by step change of the relative position of the write and read laser spots in accordance with the selected trajectory as illustrated in Fig. 3. In the *XY* coordinate frame centred at the apex of the readout spot as shown in Fig. 2, the relative position of the write spot form a 3x3 coordinate matrix as shown in fig 3a. The positioning of the write and read spots for each point of the selected trajectory was automatized with the following algorithm: (i) randomized positioning of the read spot on the device surface (ii) adjusting the focusing of the read beam, (iii) adjusting the focusing of the write beam and the spatial overlap of the write and read spots, (iv) shift of the read spot to the required position. The experimentally estimated error of the automatized positioning was 0.2 μm for all three-coordinate axis. The signal in every point of every trajectory was averaged over 10 000 measurements. Each trajectory was measured 20 times with averaging of the resulting waveform. 10 sets with six selected trajectories in each were measured.

Reviewer #2, Remark 1

The idea of wave-tronic paradigm is promising for future development neuromorphic technology. However, the authors should better explain the essence of the concept, especially in the content of the role of waves in the functioning of the biological cortex, a research area which is now booming in neuroscience.

Answer

We are grateful to the reviewer for this question, which allows us to compare the functioning principles of the developed device and the information processing by the biological cortex. We have extended the corresponding discussion in the final part of the manuscript:

Lines 296 – 311

Recent proposals indicate that wave phenomena in the brain could constitute alternative information processing, complementary to the commonly accepted mechanism of learning via developing connections between neurons [52]. For example, it has been shown that the brain can achieve selectivity function as well as stimulus detection via interference of neural waves, even if the tissue is fixed and does not accumulate information via learning [40]. In this case, the brain tissue operates as wave reservoir supporting the interaction of waves with two components – excitatory and inhibitory, reminiscent of the interacting phonons and magnons in our reservoir. With this mechanism, information processing in the visual cortex is realised via excitatory-inhibitory wave interference or mixing, even further deepening the analogy with the proposed on-chip phonon-magnon reservoir. More generally, cognitive functions could be understood in terms of spatio-temporal pattern formation [53]. Therefore, the role of standing and travelling waves in brain activities attracts significant attention [54]. The understanding of neuromorphic functions in terms of waves interaction become increasingly more important with development of quantum AI and quantum reservoir computing [55-58], where waves phenomena play the prime role. All this indicates potential and possibility for a new type of neuromorphic technology – wavetronics – where elements utilise traveling and standing waves for realisation of neuromorphic functions. Our results contribute to development of this novel technology.

Reviewer #2, Remark 2

The author use point-by-point approach to input symbols, which obviously not the best way to input the reservoir because it does not promote wave mixing. Does this imply that their reservoir is linear. Can the authors comment if/how the nonlinearity is realized in their reservoir.

Answer

Nonlinearity is an intrinsic property of the proposed reservoir, which does not depend on the writing method. The waveform generated at every point of the discrete trajectory is already a mixing of an infinite number of interacting phonon and magnon modes. They are mixed linearly due to the phase-sensitive interference of the propagating phonon modes and nonlinearly due to the nonlinear character of magnon-phonon interaction [39]. However, the contribution of magnon-phonon nonlinearity in our experiment is small, indeed, and cannot be visualised due to the low strain amplitudes of the guided phonon wavepackets. We have extended Figure 2 with the fast-Fourier transform spectra of the transient signals to emphasize the mode mixing and added more details to the description of physical mechanisms, which determine the reservoir functionality.

Lines 153 – 162

The readout waveform is sensitive to both horizontal and vertical shifts, Δx and Δy . The strong sensitivity of $M(t)$ to the relative position is governed by the properties of the propagating phonon wavepacket. Each W-mode is characterized by its individual spatial profile, frequency, velocity, and decay rate and excited by the write laser pulse with individual initial amplitude and phase. As a result of the modes' interference, the phonon wavepacket experiences continuous transformation during its propagation along the x -axis as illustrated in the Supplementary Video. Moreover, the superposition of the phonon modes varies also along the wavefront (y -axis) due to the finite sizes of the write and read laser spots. This spatial-temporal transformation imprinted into the magnon S-modes results in a unique spatial, temporal, and spectral variation of the readout. Notably, it is reminiscent of the interaction of neural waves in the visual cortex, which was recognized as one of the computation mechanisms in the brain [40].

Lines 224 – 231

The recognition becomes completely impossible if the stop band of the filter is 0-20 GHz, which also suppresses the contribution of bulk and surface phonon modes. However, cutting the frequencies above 20 GHz with preserving the main information-capable range (0-20 GHz) significantly reduces the recognition accuracy. It indicates the role of high-frequency components of the readout waveforms such as the high-order W-band harmonics, parametrically mixed first-order W-modes and broadband noise, which could play a constructive role by promoting mode mixing and enhance nonlinearity effects [43]. The separation and visualization of the specific high-frequency contributions and noise, however, are out of the scope of the present study.

Reviewer #2, Remark 3

3) Author state that the readout ANN was trained with learning rate with 0.001. Does the result depend on the learning rate?

Answer

The learning rate is a hyperparameter of the readout ANN, which is the largest magnitude at which any parameter in the network can change when the back propagation algorithm reduces the error. We have tuned this and all other hyperparameters (the number of epochs, the early stopping patience, the random seed, the number of learning layers, and the number of nodes in each of those layers) using a grid search approach to find the optimal value. The result does not depend on the learning rate, until it is too large or too small. We have added the following remark to the main text:

Lines 200 – 205

The ANN learning rate is the largest magnitude the parameters of the network can change within one epoch. It and the other ANN hyperparameters (the number of epochs, the early stopping patience, the

random seed, the number of learning layers, and the number of nodes in each of those layers) were adjusted using a grid search approach to achieve the fastest training, but their absolute values can be varied without affecting the recognition.

Reviewer #2, Remark 4

4) It is known that noise plays an important role in reservoir computing, but this discussion is missed in the paper. Could the authors to explain this better?

Answer

While our reservoir allows the identification of very noisy signals, we did not examine the noise information content. The potential role of noise is only briefly discussed in the revised manuscript (see the answer to Remark 2)

Reviewer #2, Remark 4

5) Additional relevant papers could be added in the reference list, to make the bibliography more complete.

Answer

We have extended the reference list with the following references:

Refs [10] – an overview of the physical reservoir concepts;
Refs [17-19,24] – several missed articles on the photonic, phononic and magnonic reservoir concepts.
Refs [29-32,39] – on the physical mechanisms behind the suggested reservoir' functionality.
Refs [41] – on forming the dataset for recognition by mixing.
Ref. [43] – on the role of noise.
Refs [44-47, 64] – on reservoir robustness, i.e. its temperature stability.
Ref [47-49] – on potential development of the suggested approach for reservoir computing using distributed optical and piezoelectric generation of phonon wavepackets.
Refs [53-58] – on wavetronic concepts of the biological cortex functionality.
Ref. [64] – on ANN training

Reviewer #3, Remark 1

In the introduction section (line 63) the word 'quanta' is used to refer to photons, phonons and magnons used in previous implementations using different physical waves. To me, this gives the impression that something intrinsically 'quantum' is being implemented while in fact, at least in the present proposal, the physical waves are in a 'classical' regime, being mostly its wave nature the one exploited. I think it might deserved some clarification in the current context with many quantum computing proposals.

Answer

The reviewer is right that the proposed device operates in the "classical" regime, and using quantum terms could confuse the reader. We prefer to keep the commonly used terminology, which makes the text more comprehensible, but we have modified the confusing sentences:

Lines 61 – 64

One of the first realizations was based on waves, namely gravity waves in a vessel with water to recognize spoken numbers [11]. Nowadays, wave 'reservoirs' have been implemented using different physical waves such as electromagnetic waves (photons) [12-18], elastic waves (phonons) [19,20], and spin waves (magnons) [21-24].

Lines 82 – 86

The device, shown schematically in Fig. 1b, consists of a sandwich of GaAs/AlAs semiconductor layers, which form a waveguide for high-frequency acoustic waves, i.e. phonons [28]. The waveguide is capped by a nanograting patterned in a metal ferromagnetic $\text{Fe}_{0.81}\text{Ga}_{0.19}$ layer hosting spin waves, i.e. magnons [29].

Reviewer #3, Remark 2

The authors mention that ‘the laser pulses are converted in the device to a coherent multimode phonon wave-packet’ but never explain the mechanism behind that conversion. Given that the coherence of the phonon is relevant, I suggest a brief explanation to be added.

Answer

We have added a detailed description of the excitation mechanism to the text.

Lines 99 – 113

The physical processes involved in the operation of the fabricated nanodevice are demonstrated in Fig. 2. The 100-fs pulse of the write laser, focused on the metallic ferromagnetic layer to a spot of 3- μm diameter instantaneously induces thermal stress leading to the generation of the coherent acoustic wavepacket of hypersound (up to ~ 100 GHz) frequencies [30,31]. Due to the periodic surface patterning and the corresponding spatial modulation of the optically-induced stress along the x -axis, coherent phonons acquire the wavevectors: $k_x = n 2\pi/d$, where $d = 200$ nm is the nanograting period and $n = 1,2,3\dots$ is the harmonic order [32]. The phonon wavepacket propagates along the x -axis away from the optically excited area. There are two groups of propagating phonon modes with alternative localizations. The surface modes of two polarizations, so-called Rayleigh and Sezawa waves, with the frequencies of 12.2 and 13.8 GHz ($n = 1$), respectively, are localized in the near-surface region. Their propagation is suppressed due to the scattering at the corrugated surface, and their amplitudes drastically decrease within the propagation distances of several microns. The second group of modes, so-called W-modes, are localized in the GaAs/AlAs sandwich, which has higher acoustic impedance than the impedances of the substrate and the ferromagnetic cap. The W-modes are analogue to the Lamb modes in plates and behave as waveguide modes, which can propagate on large distances protected from scattering [28,33].

Reviewer #3, Remark 3

Since there is a strong sensitivity of $M(t)$ to the characteristics of the W phonon modes, in particular initial amplitude and phase, is it to be expected that temperature changes or sample to sample variations could lead to relevant changes on the calibration of the device?

Answer

The excitation of the reservoir phonon modes occurs through a highly nonequilibrium state of the metal ferromagnetic film in the optically excited area. The amplitude and phase of the modes are determined by the amplitude and spatial profile of the instantly (within 1 ps) induced thermal stress, which itself is determined by the intensity and spatial shape of the optical excitation and the structural design. The background heating accompanying the optical excitation is much smaller than the peak electron and lattice temperatures and does not affect the readout signals. We have added a corresponding remark to the discussion of the reservoir robustness:

Lines 242 – 244

The weak temperature dependences of the elastic and magnetic parameters of the used materials around room temperature [44-46] makes the reservoir functionality also resistant to the unavoidable temperature variations due to background heating.

Reviewer #3, Remark 4

I do not understand the signals shown in the right panel of fig 3b. Why are they so clean and not noisy as the ones shown in fig3a or in supplementary fig 2?

Answer

In the main panels with the transient signals, we show the signals filtered by a bandpass Fourier filter (15-20 GHz) to remove the broadband noise and emphasize the difference of the waveforms measured for different experimental conditions. The signal processing and the following recognition were done with unfiltered “raw” signals. This clarification should have been included in the text, but was somehow missing. We have included it in the Methods section of the revised manuscript and also added “noise-free waveforms” in all the figure captions.

Lines 357 – 359

To show more explicitly the volatility of the readout waveform, the readout signals presented in Figs. 2d, 2f, 2g, and 3c were passed through a 15-20 GHz band pass filter, which eliminates the broadband noise.

Reviewer #3, Remark 5

It is my understanding that in order to calibrate the device, the writing laser is positioned on each one of the 3x3 pixels, the corresponding M_{ij} signal recorder. These signals are then superimposed according to the pattern and then the entropy, variance and skewness of it calculated. What it is a bit unclear to me is the testing procedure: are the corresponding M_{ij} signals of the intended (unknown) pattern to be recognized all add up and then the 3 parameters calculated? Or are the different combinations that defined the 'recognized shaped' calculated to see which one match better?

The device is not calibrated. The waveforms corresponding to the specific “visual shapes” are measured directly by integration of the signals over the corresponding trajectory (we have added this method, please see the reply to Remark 3 of Reviewer 1) or superimposed, as stated in your remark to create a large data set for ANN training. Each signal is processed to obtain the chosen parameters: entropy, variance, and skewness. The data set is split into training and validating sets (60% and 20% of the total dataset's size, respectively), which are used for training the algorithm. The validating set prevents the Artificial Neural Network (ANN) from overfitting, which would mean that the ANN learns the training data set. To test the resulting trained ANN, we take a testing data set (20% of the total dataset's size) and perform inference on this dataset. We pass the data through the trained ANN, and the algorithm predicts which category the pass (i.e., the combination of entropy, variance, and skewness) belongs to. It is important to note that the split of the data is non-overlapping and non-repeating. Therefore, the testing set is unseen by the algorithm, but it only holds the symbols (L, O, Minus, Plus, T, Z) that the ANN algorithm has been trained upon. The data creation, the superposition of each laser readout, must be standardized across both the training sets and the testing set, otherwise the prediction from the ANN is unreliable and inconsistent for training and testing.

The described procedure is a standard approach in machine learning and more detailed information may be found in Ref [64], which was added to the corresponding section in Methods.

Reviewer #3, Remark 6

f) What happens if the pattern that is intended to be read is not one of the 'recognized patterns', say a X?

Answer

A random waveform that does not correspond to any of the “visual shapes” selected for training the ANN will be randomly attributed to a specific shape or not recognized. We have checked that waveforms obtained by randomized drawing of trajectories do not form any clouds in the parameter space, even if the trajectories consist of the same number of discrete positions. A corresponding comment was added to the manuscript:

Lines 192 – 194

To exclude the artificial character of the parameters' distinction, we have checked that the waveforms, which do not correspond to the selected trajectories but consist of the same number of discrete positions, do not form separate clouds.

Reviewer #3, Remark 6

What about the time scale for writing? Can one use a time dependent pattern for writing? What are the limitations for that? The author speculate on that in the conclusion section, please elaborate.

Answer

We are grateful for this question, which has forced us to extend the corresponding discussion in the text. The “direct drawing” method may be considered as time-dependent input (time-dependent positioning of the

write laser on the surface) but with extremely slow time variation due to the low strain amplitude in the optically generated phonon wavepackets and, thus, long acquisition time. Increasing the input energy efficiency by two orders of magnitude will drastically accelerate the signal acquisition and, thus, processing. We have added a corresponding discussion into the text (see also the reply to Remark 1 of Reviewer #1):

Lines 258 – 273

The transmission of optical excitation to coherent phonons guided along the surface has low energy efficiency, because a vast amount of the absorbed optical energy is converted to non-coherent phonons and bulk phonon modes escaping to the substrate. Each optical pulse with an energy of 6×10^{-10} J is converted to a phonon wavepacket carrying only $\sim 10^{-16}$ J (see Methods for the details of estimations). This value is 5-orders of magnitude smaller than a single neural spike in the human brain, which estimated energy is ~ 10 pJ [8]. The averaged acoustic power involved into the signal processing is just about 10 nW. Due to these extremely low numbers, the signals are averaged over 10^{10} ‘write’ pulses/phonon wavepackets at every discrete point of the selected trajectory to achieve reasonable signal-to-noise ratio. It takes about 2 minutes for every step of the “visual shape” drawing and, thus, limits the operational speed.

Lines 287 – 296

Another direction is real-time operation with changing information patterns. Nowadays reservoir computing is considered as one of the best machine learning frameworks for temporal or sequential data processing [48]. The ability of the presented device to operate in the GHz frequency range promises high speed of neuromorphic operations. It requires, however, input methods with significantly higher energy efficiency. We consider a piezoelectric technique of generation of acoustic waves to be the most prospective in this context. Modern RF acoustic filters, which operate in billions of wireless communication devices, possess high (up to 90%) efficiency of electric to acoustic signal conversion [49]. The carrier frequency is currently limited by several GHz, but has to be extended due to the requirements of the next (6G) communication standards. Prototype devices already demonstrate an energy conversion efficiency of several per cent at frequencies above 10 GHz [50,51]. It will allow real-time operations using the proposed reservoir architecture keeping the total energy costs very low. Moreover, a two-order of magnitude increase of the strain amplitudes will drastically enhance the contribution of magnon-phonon nonlinearity [39], which is manifested only indirectly in our study.

REVIEWER COMMENTS

Reviewer #1 (Remarks to the Author):

I continue to think that this manuscript's motivation is in the wrong direction for the work presented.

I now see that the human visual system is just an analogy. But the way it is presented, a reader is likely to come away with the impression that they are making an RC that can replace the visual system, which it cannot.

Disregarding this stylistic approach, I very much disagree with their analogy. They claim that the retina and photoreceptors are the reservoir computer. I strongly disagree with this assignment. The signals generated by the retina are drastically compressed to a small number of spike trains, which are transported to the visual cortex via that optical nerve. The signals are expanded to a high-dimensional space in the visual cortex, where the information is processed. The analogy of the reservoir is the visual cortex, not the retina as they write.

My suggestion is to completely drop the very vague and potentially misleading motivation of the human visual system.

The authors did not address the fact that they need a separate artificial neural network for processing the data generated by the phonon waves. They do not include this in their energy budget calculation and this step is likely to remain even after they figure out how to engineer the "other laboratory equipment." I also worry that the post-processing artificial neural network is doing all the work. What if they just fed the images from Fig. 3b into a similar post-process network? I suspect this will work just fine and obviates the need for the phonon waves.

They also mention that they need to repeatedly send in the signal to their RC because the signals are so weak and that it takes two minutes of data collection per image. During this extended training time, they consume ~ 5 mJ of energy. There is no context for this number and it seems quite high. For example, if I just use a software reservoir computer programmed using a compiled language such as C++, the training time can be on the order of 100 microseconds on a current laptop computer. If the computer is consuming about 100 W, this translates to 10 mJ of energy consumption. So their comparison is not all that impressive. While a single operation is small, the smallness means they need a huge amount of signal averaging.

Reviewer #2 (Remarks to the Author):

The three referee reports provided very many comments to the authors. Their replies were very thoughtful, careful, and persuasive. The initial submission was very good, and this revised version is even better. I very strongly support the publication of this revised and improved version.

This is very interdisciplinary work crossing different disciplines and thus very suitable for Nature Communications. It is difficult to please everybody, especially when the readers belong to very different disciplines. Still, this work has done a very good job in presenting a very interesting and insightful study, of interest to researchers working in very different fields.

Incidentally, several claims by referee 1 are puzzling, because the original text did not write several claims that referee 1 assigned to the initial manuscript. It is not clear why that report misread the paper so much, as if it had read a different work.

Reviewer #3 (Remarks to the Author):

I find the authors' reply to the referees' reports overall satisfying, providing a clear response to all comments and criticisms. I also think that the authors have substantially improved the manuscript in the process by providing additional information. I therefore support its publication in Nat. Comm.

We thank all reviewers for their careful work with our manuscript. We are pleased that our responses and modifications of the manuscript were appreciated by Reviewers 2 and 3. We are also grateful to Reviewer 1 for the additional questions and for pointing out the aspects of the manuscript which could be further improved.

Below, we give detailed answers to the reviewer's remarks and highlight the corresponding changes made in the manuscript.

Remark 1.

I continue to think that this manuscript's motivation is in the wrong direction for the work presented.

I now see that the human visual system is just an analogy. But the way it is presented, a reader is likely to come away with the impression that they are making an RC that can replace the visual system, which it cannot.

Disregarding this stylistic approach, I very much disagree with their analogy. They claim that the retina and photoreceptors are the reservoir computer. I strongly disagree with this assignment. The signals generated by the retina are drastically compressed to a small number of spike trains, which are transported to the visual cortex via that optical nerve. The signals are expanded to a high-dimensional space in the visual cortex, where the information is processed. The analogy of the reservoir is the visual cortex, not the retina as they write.

Answer

We are grateful to the reviewer for this detailed description. Indeed, human vision is a bright real-life example of reservoir computing. This is why it is so often used to illustrate the reservoir computing concept. From the feedback from colleagues, we know that it helps to explain reservoir computing to non-specialists. Moreover, the similarity of our concept with the operation principles of a biological cortex was also pointed out by Reviewer 2, who asked us to extend the corresponding discussion in the manuscript (please see the previous response; Reviewer 2, Remark 1).

Thus, we would like to keep Figure 1. We are grateful to the reviewer for the corrections and have made the following modifications to the manuscript.

Figure 1a.

The scheme and the labels have been modified following the reviewer's comment.

Lines 50-55

As one of many nature prototypes of RC, one could consider the human vision schematically illustrated in Fig. 1a. The visual information passed through the cornea and focused by the lens is converted by the retina photoreceptors into electrical signals (the neural pulses) that are transmitted by the optic nerve and mixed in a high-dimensional space of the visual cortex for the subsequent recognition. The complex photochemical transformation (phototransduction) by the 10^8 photoreceptors allows our brain to recognize objects, distinguish tiny differences between them, and detect thereby their motion.

Lines 92-96

We may compare this procedure with handwriting and conceptualize the reservoir operations as recognition of a handwritten character. Thus, in analogy with the human vision (Fig. 1d), we may consider the patterned surface as the retina, which converts optical input to neural pulses (coherent phonons), processed by a multidimensional reservoir space (phonon-magnon mixture) to generate a readout recognized by the brain, namely the ANN.

Because it is clearly stated in the text that human vision is just one of many examples of real-life reservoir computing, and neither in "Introduction" nor in "Perspectives" we claim that our concept can replace or mimic human vision, we are sure that the reader will not be misled.

Remark 2.

The authors did not address the fact that they need a separate artificial neural network for processing the data generated by the phonon waves. They do not include this in their energy budget calculation and this step is likely to remain even after they figure out how to engineer the "other laboratory equipment."

Answer

We are grateful for this comment, which has forced us to make this part of the manuscript more transparent for the reader.

The presence of the ANN and its role in the recognition are described in detail in the manuscript (lines 196-221). This chosen scheme corresponds to the general definition of the physical reservoir concept:

*A reservoir computing system consists of a reservoir for mapping inputs into a high-dimensional space and a readout for pattern analysis from the high-dimensional states in the reservoir. The reservoir is fixed and only the readout is trained with a simple method such as linear regression and classification. [G. Tanaka et al., Recent advances in physical reservoir computing: A review. *Neural Networks* **115**, 100 (2019)]*

and its typical architecture:

Figure 1 ... and (c) a reservoir made of untrained connections. For RC, only the connection between readout layer and output layer is trained, typically with a linear regression. [M. Cucchi et al., Hands-on reservoir computing: a tutorial for practical implementation. *Neuromorph. Comput. Eng.* **2, 032002 (2022)].**

In our system, the reservoir readout is only three real numbers. Nevertheless, the “visual shapes” are well separated in such a simple space (see Figs. 3d and S3) and can be classified using various algorithms. We use a simplified ANN, which serves to just accelerate the training of the readout weights. This approach is often used in reservoir concepts (see for example Moon et al. *Nat. Electronics* **2** 480 (2019)] or Zhnag et al. *Nat. Commun.* **13**, 6590 (2022)).

The presented energy budget estimates only the energy costs for the signal processing by the reservoir (please see the answer to Remark 4) and does include the ANN, which energy costs are negligibly low.

To better explain our choice of the realised architecture we have made the following changes in the manuscript:

Figure 3a and the inset in Figure 4.

We have added the label “reservoir readout.”

Figure 3 and Methods sections (Line 396).

We have substituted the potentially confusing term “Signal processing” with “Waveform characterization”, which is a more appropriate name for calculating the basic parameters of the waveforms.

Lines 196-199

Since the reservoir demonstrates high selectivity in filtering symbols, that are well separated in the parameter space, various algorithms for the symbol classification can be realized. We have chosen the conventional physical reservoir approach [5,10] with training the readouts. To accelerate the

training process, we use a simple ANN forming a feedforward Multi-Layered Perceptron with a back-propagation algorithm [42].

Remark 3

I also worry that the post-processing artificial neural network is doing all the work. What if they just fed the images from Fig. 3b into a similar post-process network? I suspect this will work just fine and obviates the need for the phonon waves.

Answer

The images in Fig. 3a are idealized and artificially constructed in a graphical redactor (please see the figure caption). They are obtained by superimposing the images of the write laser spot captured by a CMOS camera using 500x-magnification (Methods, lines 258-261). This approach would neither allow an on-chip implementation nor provide a sub-wavelength (100-nm) spatial resolution. It would also require a significantly more complex ANN, which should have at least 9 inputs, with questionable recognition results. Thus, this will not work.

Remark 4.

They also mention that they need to repeatedly send in the signal to their RC because the signals are so weak and that it takes two minutes of data collection per image. During this extended training time, they consume ~ 5 mJ of energy. There is no context for this number and it seems quite high. For example, if I just use a software reservoir computer programmed using a compiled language such as C++, the training time can be on the order of 100 microseconds on a current laptop computer. If the computer is consuming about 100 W, this translates to 10 mJ of energy consumption. So their comparison is not all that impressive. While a single operation is small, the smallness means they need a huge amount of signal averaging.

With all due respect, we find this comparison not very accurate. Either the ANN in our experiment or the speculative “software reservoir computer” suggested by the reviewer (it is unclear how the 100- μ s training time was obtained) operate with already digitized inputs. A physical reservoir operates with a physical signal (e.g., electromagnetic, spin, or acoustic wave) and processes it. Thus, the estimated energy costs should be compared with typical energy costs of electronic devices operating with signals of the same type. We have extended the text accordingly:

Lines 270-274

However, the total energy costs remain extremely low: the single symbol drawing costs about 5 μ J and the total acoustic energy, which would be spent for the drawing and processing of the symbols in the whole training process, is less than 5 mJ. For comparison, it is two-order of magnitude less than the acoustics energy transmitted by a RF acoustic filter during 1-second operation of a modern wireless communication device [47].

The reasons for the noisy signals in our experiment, i.e. the low efficiency of the optoacoustic conversion, and the ways to significantly improve the signal-to-noise ratio and decrease the acquisition times are already discussed in the manuscript (Lines 262-264 and 292-300).